∂ | **Open Peer Review** | Clinical Microbiology | Research Article

# Gut microbiota-associated non-cholesterol sterol dysregulation modulates immune reconstitution during antiretroviral therapy in people living with HIV

Jingying Pan,[1] Xuebin Tian,[1,2] Kai Wu,[1] Jia Ji,[1] Mingqing Dong,[3] Ting Sun,[1] Dan Lv,[4] Peng Yao,[3] Longxian Lv,[1,2] Hangping Yao[1,2]

**ABSTRACT** Non-cholesterol sterol metabolism plays a crucial role in immune regulation. However, the non-cholesterol sterol profiles, its association with gut dysbiosis, and its impact on the CD4$^+$ T cell recovery in people living with HIV (PLWH) are yet to be elucidated. In this study, we recruited 37 PLWH and 50 healthy controls to characterize non-cholesterol sterol profiles and gut microbiota composition using targeted liquid chromatography–mass spectrometry and metagenomic analysis. Correlations between sterol profiles and immune cell subsets were assessed. *In vitro* peripheral blood mononuclear cell (PBMC) model was used to validate key findings. We identified a distinct dysregulation of non-cholesterol sterol metabolism in PLWH, characterized by elevated levels of cholesterol precursors and metabolites and depleted levels of plant sterols, which were linked to gut dysbiosis. Our study results highlighted *Oscillibacter* spp. as the key regulator of sterol metabolism. Specifically, plant sterols (e.g., brassicasterol and campesterol) were found to be associated with impaired CD4$^+$ T cell recovery during antiretroviral therapy (ART). These findings were validated using *ex vivo* PBMC models, which revealed that brassicasterol stimulates T cell abnormal activation and pro-inflammatory cytokine release, whereas lathosterol dampens immune activation and inflammation. In summary, our study highlights the interplay between gut dysbiosis and sterol dysregulation in PLWH, demonstrating that higher brassicasterol levels impair immune recovery post-ART by promoting CD4$^+$ T cell hyperactivation. Hence, targeting microbial sterol metabolism—through *Oscillibacter* spp. enrichment or plant sterol modulation—may offer novel therapeutic strategies to optimize ART outcomes by balancing immune activation and resolution.

**IMPORTANCE** This study is the first to integrate non-cholesterol sterol profiling with gut microbiota analysis in people living with HIV (PLWH), uncovering a unique sterol dysregulation characterized by elevated cholesterol precursors and depleted plant sterols in this population. We demonstrate that *Oscillibacter* spp. were associated with these metabolic shifts and that specific sterols differentially affect immune recovery: plant sterols such as brassicasterol impede CD4$^+$ T cell restoration by promoting hyperactivation, whereas the cholesterol derivative lathosterol mitigates inflammation and supports immune reconstitution. These insights reveal novel microbiome–sterol interactions that can be leveraged to develop targeted microbiome- and sterol-based interventions aimed at enhancing antiretroviral therapy efficacy and long-term immune health in PLWH.

**KEYWORDS** human immunodeficiency virus, gut microbiota, non-cholesterol sterols, immunological responders, immunological non-responders

**Peer Reviewers** Ajay Kumar Sahu, Bangalore University, Bhubaneswar, Odisha, India; Shahid Siddhik Attar, ADT's Shardabai Pawar Mahila Arts, Science, and Commerce College Baramati, Pune, India

Address correspondence to Longxian Lv, lvlongxian@zju.edu.cn, or Hangping Yao, yaohangping@zju.edu.cn.

Jingying Pan and Xuebin Tian contributed equally to this article. Author order was determined based on contribution to this article.

The authors declare no conflict of interest.

See the funding table on p. 16.

As of the end of 2023, over 39.9 million people globally are living with the human immunodeficiency virus (HIV). In addition, more than 1 million new cases are reported annually. Therefore, HIV is a significant global public health challenge (1). While combination antiretroviral therapy (cART) has revolutionized HIV management by suppressing viral replication and reducing AIDS-related morbidity and mortality (2), significant health disparities still persist. Despite advancements in ART, the median survival time for people living with HIV (PLWH) aged 50 and above remains lower than that of their HIV-negative counterparts (3).

Chronic immune activation and systemic inflammation—hallmarks of HIV infection that persist despite suppressive ART—are major contributors to non-AIDS-related morbidity and mortality. Central to this process are CD4$^+$ T cells, which are not only the primary targets of HIV but also play a critical role in orchestrating adaptive immune responses. Although quiescent CD4$^+$ T cells express intrinsic post-entry restriction factors such as MCPIP1 that inhibit viral replication (4), HIV-induced immune activation expands the pool of activated CD4$^+$ T cells, accelerates their turnover, and ultimately leads to immune exhaustion and disease progression (5, 6). Despite long-term ART and sustained viral suppression, approximately 10–40% of PLWH fail to achieve adequate immune reconstitution and are classified as immunological non-responders (INRs), who experience poorer clinical outcomes compared to immunological responders (IRs) (7, 8).

Concurrently, PLWH face a substantially elevated risk of cardiovascular diseases (CVDs), which have become the leading cause of non-AIDS-related mortality in this population (9, 10). This increased risk is largely driven by a high prevalence of metabolic syndrome components—including dyslipidemia, central obesity, insulin resistance, and hypertension—that act synergistically to exacerbate cardiovascular burden (3, 11–13). Among these risk factors, high cholesterol absorption has emerged as a critical contributor to CVDs, underscoring its importance in the pathophysiology of cardiovascular morbidity in PLWH (14). Non-cholesterol sterols, including plant sterols (e.g., sitosterol and campesterol), cholesterol precursors, and cholesterol metabolites, play a crucial role in regulating cholesterol absorption and synthesis (15). Cholesterol precursors, in particular, serve as biomarkers for endogenous cholesterol synthesis and are instrumental in diagnosing disorders related to cholesterol biosynthesis (16). Plant sterols are widely recognized as surrogate markers for assessing gastrointestinal cholesterol absorption (17). Beyond their role in cholesterol metabolism, non-cholesterol sterols are essential intermediate metabolites involved in numerous biological processes, particularly those associated with inflammatory diseases (18–20). Growing evidence suggests a link between innate immune signaling pathways and the regulation of sterol metabolism (21–26). Non-cholesterol sterols regulate membrane fluidity by counteracting cholesterol-induced rigidity and exhibit dual inflammatory effects, enhancing pro-inflammatory cytokines while suppressing IL-1β (27). They amplify macrophage responses via AP-1 and bridge lipid metabolism with immune regulation, as shown in animal models (28). Understanding the interplay between these metabolic pathways may provide valuable insights into the mechanisms underlying the increased CVD risk and unfavorable CD4$^+$ T cell reconstitution in PLWH and inform targeted therapeutic strategies.

Recent studies have shown that HIV infection causes the strongest disease-associated dysbiosis in gut microbiota, leading to an overabundance of pro-inflammatory and potentially pathogenic bacterial populations, genes, and metabolic pathways. In addition, PLWH are more susceptible to co-infections with other pathogens—including mpox, sexually transmitted infections, hepatitis B virus, hepatitis C virus, and *Treponema pallidum*—which may further exacerbate immune dysfunction and complicate clinical management (29–33). Notably, the disruption of gut mucosal integrity and microbial composition following HIV infection may itself enhance host vulnerability to secondary infections (30). Compared to HIV monoinfection, co-infection is associated with more pronounced gut microbiota alterations, which may contribute to systemic immune activation and HIV-associated neuropsychiatric complications (32). Furthermore, shifts in gut microbiota composition have been closely linked to immune reconstitution

during ART (34–38). The gut microbiome functions as a complex bioreactor, synthesizing and breaking down neurotransmitters, short-chain fatty acids (SCFAs), lipids, vitamins, amino acids, and other metabolites (39). Several of these gut microbiota metabolites, including bile acids and SCFAs, are crucial for immune system modulation (40–42). Recent evidence suggests that microbiota-derived SCFAs, particularly propionate, could be targeted to modulate disease progression and inflammation-related comorbidities in virally suppressed individuals with HIV (43). Additionally, gut microbiota-derived lactate-producing bacteria have been reported to enhance CD4[+] T cell recovery during ART in PLWH (44).

Some recent studies have shown that apart from the traditional metabolites, gut microbiota can also metabolize cholesterol (45–48). Uncultured *Eubacterium* species carrying the ismA gene, which encodes for the IsmA enzyme responsible for converting cholesterol to coprostanol, have been found to be significantly associated with lower levels of both stool and plasma cholesterol (46). Additionally, recent research has shown that the sulfotransferase enzyme from *Bacteroides thetaiotaomicron* is involved in the production of cholesterol sulfate, which correlates with the elevated levels of plasma cholesterol and cholesterol sulfate (47, 48). A higher relative abundance of *Oscillibacter* species correlates with reduced cholesterol levels in blood and stool. The presence of genes encoding cholesterol-related proteins (IsmA and CgT) in multiple species from *Oscillibacter* also supports their role in cholesterol reduction (45). Further investigations have revealed potential cholesterol-metabolizing activity in several gut microbiota species, including *Bifidobacterium*, *Enterococcus*, *Lactobacillus*, and *Parabacteroides* (45).

We are interested in elucidating the sterols and cholesterol metabolism status in PLWH and the relationship between sterol and cholesterol metabolism and gut microbiota dysbiosis in PLWH, and assess how these factors influence immune recovery. This is a prospective cross-sectional cohort study, in which blood and stool samples are collected from the PLWH undergoing regular ART treatment as well as from healthy individuals. It analyzes the gut microbiota composition and serum sterol levels of both the PLWH and healthy individuals to assess their correlations. Additionally, it examines the impact of abnormal serum sterol distribution in PLWH on immune reconstitution post-ART. The findings are validated using an *in vitro* cell model. Integration of these multi-faceted analyses allowed us to define how sterol–microbiota interactions shape the immune system and ultimately determine the trajectory of CD4[+] T cell recovery in PLWH.

## MATERIALS AND METHODS

### Study population

Between November 2020 and December 2022, we recruited 37 PLWH from the HIV clinic of the First Affiliated Hospital, College of Medicine, Zhejiang University and the Disease Control and Prevention Center of Zhejiang Province, along with 50 healthy controls (HCs) from the Health Testing Center of the hospital. All PLWH had initiated ART at the chronic phase of infection. None of the participants were on anticholesterol medications before or during the study. Individuals who had taken probiotics, antibiotics, or both within 4 weeks prior to enrollment were excluded. The study followed the Helsinki Declaration. The Institutional Review Committee of the First Affiliated Hospital, College of Medicine, Zhejiang University, had approved the protocols (ethical number IIT20230314B). All participants provided written informed consent.

### Sample collection

In total, 87 fresh stool samples and 87 serum samples were collected from 37 PLWH and 50 healthy individuals, along with 37 whole blood samples from PLWH. The stool samples were immediately quick-frozen in liquid nitrogen upon collection and then stored at −80°C until analysis. The serum samples were aliquoted into 100 µL portions

upon collection to prevent repeated freeze–thaw cycles and reserved for later serological analysis. Finally, all aliquoted serum samples were stored at −80°C and tested within 3 months of receipt. The whole blood samples (37 in total, 8 mL each), collected from the PLWH for routine hematological analysis, were retrieved. Immediately following collection, flow cytometry was used to enumerate the lymphocyte subpopulations in the samples obtained. Additionally, peripheral blood mononuclear cells (PBMCs) from the PLWH enrolled in the study were isolated following established protocols (49) and stored at −80°C for the subsequent analysis.

## Liquid chromatography–mass spectrometry (LC–MS)

Serum samples (50 µL) were first mixed with 150 µL of chilled methanol for protein precipitation, and then centrifuged at 12,000 rpm and 4°C for 10 min. The obtained supernatants were transferred to new tubes, evaporated to dryness, and derivatized with 200 µL of pyridine solution (containing PA [8 mg], MNBA [22 mg], and DMAP [4 mg]) at 80°C for 60 min. The reaction was stopped using 200 µL of water, and the derivative was extracted using 200 µL of methyl tert-butyl ether (MTBE). After vortex-mixing, the MTBE layer was transferred, and the top layer was re-extracted using another 200 µL of MTBE. The extraction was dried and re-dissolved in 200 µL of acetonitrile, and then centrifuged at 3,000 rpm for 15 min. A 100 µL aliquot was used for UPLC–MS/MS analysis. Next, 10 µL of the sample was diluted 2,000 times with acetonitrile for cholesterol detection.

UPLC–MS/MS was performed by using an Agilent 6470 triple quadrupole mass spectrometer coupled to an Agilent 1290 UPLC system equipped with an electrospray ionization (ESI) source. The extract (1 µL) was injected onto a ZORBAX Eclipse Plus C18 column. A 7-min gradient (90–100% B) was used for hydroxysteroids, while a 17-min gradient was employed for steroids. Water with 0.1% formic acid (A) and water with acetonitrile and 0.1% formic acid (B) were used as solvents. ESI was conducted in the positive mode, with specific instrument settings for nebulizer pressure, gas temperatures, and capillary voltage. Then, they were quantified using multiple reaction monitoring.

## DNA extraction, library construction, and metagenomic analysis

Metagenomic sequencing and subsequent analyses were performed by OE Biotech (Shanghai, China), as previously described (50). Briefly, stool samples were collected and immediately processed for genomic DNA extraction using the QIAamp Fast DNA Stool Mini Kit (Qiagen, Germany), following the manufacturer's instructions. The NanoDrop 2000 spectrophotometer (Thermo Fisher Scientific, USA) was used to assess the DNA concentration and purity, with absorbance ratios at 260/280 nm for evaluating DNA purity. Agarose gel electrophoresis was employed to confirm the integrity of the extracted DNA and rule out any degradation. Following extraction, the DNA was further purified to remove any contaminants. Metagenomic shotgun libraries were constructed using the TruSeq DNA Sample Preparation Kit (Illumina, USA), which yielded an insert size of approximately 350 bp. The library construction process involved DNA fragmentation, end-repair, A-tailing, and adapter ligation. Next, PCR amplification was performed to enrich the library. The resulting PCR product was quantified using a fluorometric assay to ensure the optimal concentration of the library. The final libraries were sequenced on the Illumina NovaSeq 6000 system (Illumina, USA) using a 150 bp paired-end sequencing approach. All 87 samples passed the quality control criteria and were included in the analysis, ensuring a robust data set for downstream bioinformatic processing. Raw sequencing data in the FASTQ format was preprocessed using fastp, a tool for fast and accurate quality trimming, which removed low-quality reads and adapter sequences. To eliminate host DNA contamination, the reads were aligned to the reference genome using bbmap, which ensured the exclusion of non-target sequences. Valid reads were processed with MEGAHIT for quality control and Prodigal for ORF prediction. Gene sets were constructed using MMSeqs2, and gene abundances were quantified with the Salmon tool. DIAMOND was used to annotate amino acid sequences obtained from NCBI NR, KEGG, COG, SWISS-PROT, and GO databases. Species taxonomy and abundance

were determined using the NR Library. Statistical analysis was performed at different taxonomic levels.

## Lymphocyte subset analysis by flow cytometry

Lymphocyte subsets in the PLWH were quantified using flow cytometry. Briefly, 100 µL of whole blood was incubated with Multitest 6-color TBNK reagent (BD Biosciences, San Jose, CA, USA), which includes a combination of fluorescently labeled monoclonal antibodies designed to identify key immune cell populations. The reagent mix was prepared and used in accordance with the manufacturer's instructions to ensure optimal staining efficiency. Red blood cells were lysed using BD FACS lysing solution, and samples were analyzed on a Navios flow cytometer (Beckman Coulter). The lymphocyte populations were gated as follows: T cells ($CD3^+CD19^-$), B cells ($CD3^-CD19^+$), and NK cells ($CD16/CD56^+$). $CD4^+$ T cells ($CD4^+CD3^+$) and $CD8^+$ T cells ($CD3^+CD8^+$) were further distinguished within the T cell population.

## Cell counting kit-8 (CCK-8) assay

The PBMCs were isolated from whole blood using a density gradient centrifugation method. The PBMCs were then resuspended in an RPMI 1640 culture medium supplemented with 10% fetal bovine serum and 1% penicillin–streptomycin. A suspension comprising the culture medium (100 µL) containing $1 \times 10^5$ PBMCs was seeded in each well of a 96-well flat-bottom plate. The cells were treated with various concentrations of cholesterol, lathosterol, brassicasterol, and campesterol, as well as a PBS control. All treatments were performed in quadruplicate. The cells were cultured for 48 h at 37°C in a humidified incubator with 5% $CO_2$. Then, 10 µL of the CCK-8 solution (MCE, Monmouth Junction, NJ, USA, Cat. no. HY-K0301) was added to each well, followed by incubation at 37°C for an additional 2 h. The absorbance of each well was measured at 450 nm ($OD_{450}$). The average values were calculated and the survival rate curves were plotted based on the results obtained.

## Cell culture and cytokine analysis

PBMCs were cultured in the RPMI 1640 medium in six-well plates for 48 h. The cells were treated with varying concentrations of cholesterol, lathosterol, brassicasterol, and campesterol, along with a PBS control. After incubation, the cell culture supernatants were collected. Cytokine levels in the supernatants were quantified using the V-PLEX Proinflammatory Panel 1 (mouse) Kit (Meso Scale Diagnostics, Cat. No. K15048D), following the manufacturer's instructions (51).

## Flow cytometry analysis of treated PBMCs

The PBMCs were treated with cholesterol, lathosterol, brassicasterol, and campesterol at varying concentrations, alongside a PBS control, for 48 h. The cells were then collected for flow cytometry analysis. Fixable Viability Stain 510 (BD Biosciences) with a dilution of 1:1,000 was used to identify viable cells. Following viability staining, the cells were washed with a fluorescence-activated cell sorting (FACS) buffer (BD Biosciences) and incubated at 4°C for 30 min with a panel of fluorescently conjugated antibodies, each diluted to 1:100. The antibody panel included anti-human CD3, CD4, CD8, TCR γδ, CD45RA, CD62L, HLA-DR, CD28, CD57, and CD279 (all from BD Biosciences). After staining, the cells were rewashed and resuspended in 200 µL of FACS buffer. The CytoFLEX LX flow cytometer (Beckman Coulter, Inc., Brea, CA, USA) was used to analyze the samples. FlowJo software version 10.10.0 (Treestar Inc., Ashland, OR, USA) was used to process the data.

## Statistical analysis

Clinical characteristics of the two groups (PLWH and HCs) were compared using appropriate statistical tests. The continuous variables were compared using the Wilcoxon

rank-sum test. For categorical variables, the Fisher's exact test was employed. To assess the alpha diversity of microbiota, we calculated the ACE, Simpson, and Shannon indices and compared these diversity metrics using the Wilcoxon rank-sum test. To identify differential metabolites between the two groups, orthogonal partial least squares discriminant analysis (OPLS-DA) was performed. Microbiome multivariable associations with linear models (MaAsLin2, version 2) were used to analyze the differential abundance of KEGG level 3 pathways. To assess the significance of continuous variables between groups, we again applied the Wilcoxon rank-sum test, which is appropriate for non-normally distributed data. For multiple comparisons, we used the Benjamini–Hochberg procedure to adjust for the false discovery rate (FDR), which helps control the type I error rate when multiple statistical tests are conducted. The results were considered statistically significant if the $P$ value or FDR-corrected $P$ value was less than 0.05. Correlation between continuous variables, including heatmap between the relative abundance of *Oscillibacter* species, serum sterol levels, KEGG level 3 pathway, and immune cell subsets, was analyzed using the Spearman correlation test. To evaluate the predictive performance of the identified biomarkers, receiver operating characteristic (ROC) curves were constructed. The area under the ROC curve (AUC) was calculated to quantify the diagnostic ability of the biomarkers. All statistical analyses were performed using GraphPad Prism (version 7.0) and SPSS (version 23.0). Correlation heatmaps and associated figures were generated using the OmicStudio platform (https://www.omicstudio.cn).

## RESULTS

### Non-cholesterol sterols are dysregulated in PLWH

We recruited 37 PLWH and 50 HCs who were relatively matched for sex, smoking status, and age. Their demographics, clinical characteristics, and stool and serum collection schedules are shown in Fig. 1A. The characteristics of the participants are listed in Table S1. Our analysis revealed significant differences between the biochemical profiles of HCs and PLWH. Specifically, PLWH demonstrated lower levels of total bilirubin, indirect bilirubin, albumin/globulin ratio, albumin, and high-density lipoprotein cholesterol (HDL-C) compared to those in HCs. Conversely, PLWH had significantly higher levels of globulin and triglycerides (TGs) than those in HCs.

The cholesterol synthesis pathways are illustrated in Fig. 1B. Cholesterol is metabolized through three major pathways: steroid degradation, steroid hormone biosynthesis, and primary bile acid biosynthesis. In plants, the pathway for brassinosteroid biosynthesis, which is not present in humans, competes with that for cholesterol synthesis for the shared precursor—squalene. In addition, plant sterols, such as campesterol, are not synthesized in humans. Such sterols can only be absorbed from food through the NPC1L1 protein in the small intestine (15).

To assess the levels of non-cholesterol sterols and cholesterol in PLWH, we generated targeted metabolomic profiles using serum samples from both PLWH and HCs with the help of LC-MS. Our analysis revealed that PLWH had significantly higher levels of cholesterol precursors, such as lanosterol, and cholesterol metabolites, including cholestanol, 4β-hydroxycholesterol, 27-hydroxycholesterol, and 25-hydroxycholesterol, than those in HCs (Fig. 1C). Conversely, the levels of plant sterols, such as brassicasterol and ergosterol, were markedly lower in PLWH than they were in HCs. We utilized OPLS modeling to evaluate the differences between PLWH and HCs. Our analysis identified significant variations in the levels of non-cholesterol sterols and cholesterol between the two groups (Fig. 1D).

### Gut microbiota are involved in cholesterol metabolism disorder in PLWH

Accumulating evidence reveals the cholesterol-metabolizing capabilities of the gut microbiota (45). To investigate whether gut microbiota change in PLWH is associated with cholesterol metabolism disorder in PLWH (52), we reanalyzed the metagenomic

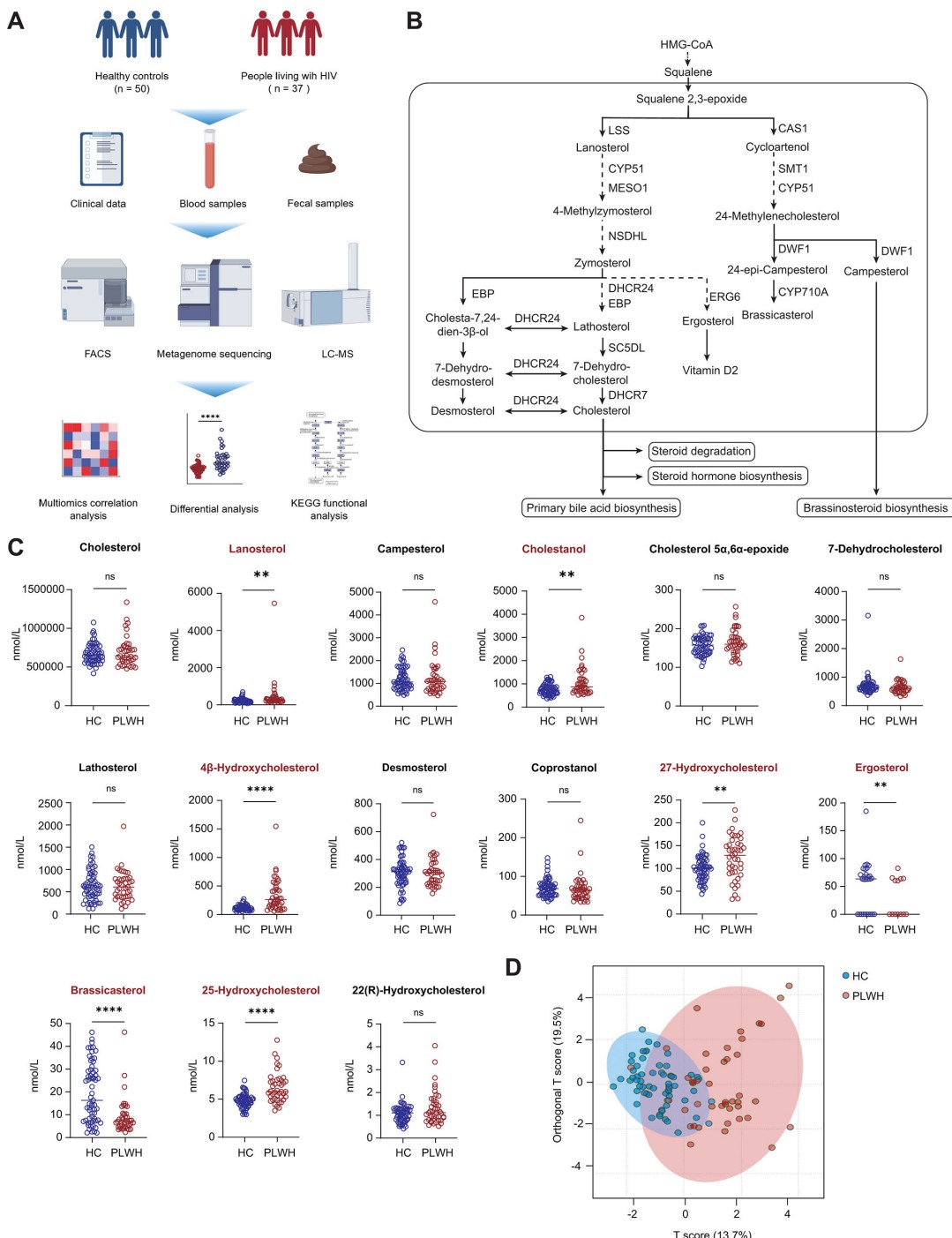

**FIG 1** Gut microbiota-driven dysregulation of non-cholesterol sterol metabolism in people living with HIV (PLWH). (A) Study design and sample collection workflow. Serum and stool samples were obtained from PLWH ($n = 37$) and healthy controls (HCs, $n = 50$). (B) Schematic of cholesterol biosynthesis. Highlighted metabolites (red) denote key intermediates quantified in this study. (C) Serum levels of non-cholesterol sterols and cholesterol in PLWH versus HC. Data presented as median ± IQR; ****$P < 0.001$, **$P < 0.01$ by Mann–Whitney $U$-test. (D) Orthogonal partial least squares-discriminant analysis (OPLS-DA) S-plot analysis of sterol profiles differentiating PLWH (red) and HC (blue).

data from our previous study (50) for both PLWH and HC groups. The species richness and diversity were found to be significantly lower in PLWH (Fig. S1). The results showed that the dominant microbial taxa in PLWH were different from those in HCs. At the genus level, Wilcoxon rank-sum testing with FDR correction ($P < 0.05$) identified 659

genera exhibiting significant compositional differences between the groups. A heatmap illustrating the relative abundance patterns of the top 30 differentially abundant genera is shown in Fig. S2. Consistent with the established literature, our study also demonstrated that SCFA-producing genera, which are epidemiologically linked to reduced risks of obesity, diabetes mellitus, and neoplastic diseases—such as *Roseburia*, *Eubacterium*, *Lachnospira*, *Alistipes*, and *Oscillibacter* (43, 52)—were significantly depleted in the PLWH cohorts (Fig. S2). To elucidate the host–microbiota metabolic crosstalk, we performed Spearman's correlation analyses between 659 differential genera identified between PLWH and HC and serum levels of non-cholesterol sterols and cholesterol. We observed significant associations ($|\rho| > 0.4$, FDR-adjusted $P < 0.05$) among these variables (Fig. 2A). Gut microbiota exhibited the strongest correlations with 4β-hydroxycholesterol, 27-hydroxycholesterol, and 25-hydroxycholesterol as well as with brassicasterol. Specifically, 4β-hydroxycholesterol, 27-hydroxycholesterol, and 25-hydroxycholesterol showed predominantly negative associations with the majority of bacterial genera, while brassicasterol primarily demonstrated positive correlations (Fig. 2A). Several bacterial genera demonstrated multisterol interactivity, with *Anaerofilum*, *Leucothrix*, *Faecalibacterium*, *Phocea*, *Anaeromassilibacillus*, *Dehalobacter*, and *Oscillibacter* exhibiting significant associations (with more than two non-cholesterol sterol species). Notably, among the genera that showed a strong association with cholesterol metabolism, *Oscillibacter* reportedly demonstrated the ability to metabolize cholesterol and has been experimentally validated and epidemiologically linked to hypocholesterolemic phenotypes (45). Fourteen *Oscillibacter* species-level assemblies (metagenomic species pangenomes [MSPs]) were identified in the gut microbiota of both PLWH and HC groups. All these 14 species had a significantly lower abundance in PLWH than that in HCs (Fig. 2B). We used Spearman's correlation analysis to ascertain the associations among non-cholesterol sterols and *Oscillibacter* MSPs. The results showed that the relative abundance of *Oscillibacter* MSPs is negatively correlated with the cholestanol, 4β-hydroxycholesterol, 27-hydroxycholesterol, and 25-hydroxycholesterol levels and positively with the brassicasterol level (Fig. 2C).

Building on these findings, we further examined the associations between the top 30 differential genera identified between PLWH and HC and host clinical parameters. Twelve genera were found to be positively associated with HDL-C levels and negatively associated with TG levels, including SCFA-producing bacteria such as *Faecalibacterium* and *Roseburia* (53). Notably, *Oscillibacter* also showed a similar pattern of association with HDL-C and TG levels. These observations provide additional support for the hypothesis that *Oscillibacter* may play a role in modulating host cholesterol metabolism (Fig. S3).

To further assess and compare the functional potential of the intestinal microbiota between PLWH and HC groups, we utilized the PICRUSt2 software tool to identify the KEGG Orthology (KO) level 3 pathways related to cholesterol metabolism (Fig. 2D). Our analysis revealed that the abundance of microbial pathways involved in the degradation of steroids and biosynthesis of steroid hormones was significantly lower in PLWH than they were in HCs. In contrast, steroid biosynthesis pathways were more abundant in their microbiota. No significant differences were observed between the primary bile acid biosynthesis pathways of the two groups. The reduced abundance of cholesterol metabolism-related pathways aligns with the depletion of cholesterol-metabolizing bacteria observed in PLWH microbiota samples. This result suggests the occurrence of impaired cholesterol metabolism in the gut, potentially leading to increased absorption of local cholesterol into the bloodstream and elevated levels of cholesterol metabolites in the blood in PLWH.

## Serum non-cholesterol sterol levels are correlated with favorable recovery of CD4+ T cells

Non-cholesterol sterols may influence immune function. To investigate whether gut microbiota-associated metabolites could impact post-ART CD4+ T cell recovery in PLWH, we compared the levels of 14 *Oscillibacter* MSPs, serum non-cholesterol sterols, and

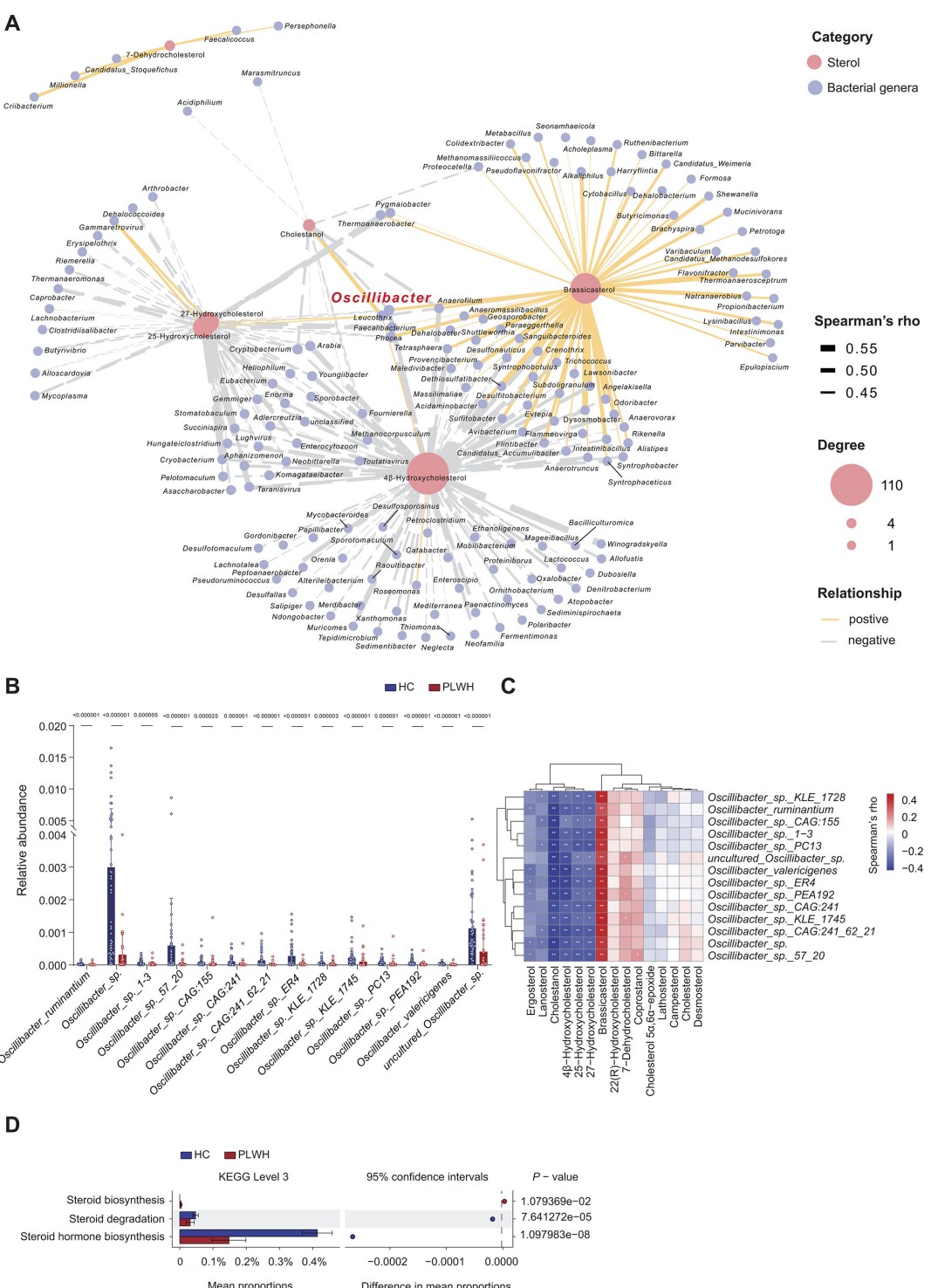

**FIG 2** Gut microbiome signatures linked to non-cholesterol sterol dysregulation in PLWH. (A) Distinct microbes interacted with non-cholesterol sterol and cholesterol levels in participants. Line thickness represents the value of the Spearman correlation coefficient. Red lines represent positive association, while blue lines represent negative association. Red circles represent the microbes, and blue circles represent non-cholesterol sterol. The sizes of the circles indicate the association degree. Only Spearman's ρ > 0.4 (P < 0.5, Benjamini–Hochberg corrected) is shown. (B) Comparative relative abundance of *Oscillibacter* species between PLWH and HC (median ± IQR; Mann–Whitney U-test). (C) Spearman's correlation heatmap between the relative abundance of *Oscillibacter* species and serum sterol levels. Color intensity reflects Spearman's ρ (q value < 0.2, Benjamini–Hochberg corrected); asterisks denote significant correlations (*P < 0.05, **P < 0.01). (D) PICRUSt2-predicted functional divergence in gut microbiota. KEGG pathways (level 3) with significant differential abundance (Welch's t-test, FDR < 0.1) are shown.

cholesterol between INRs and IRs. We defined IRs as PLWH whose CD4$^+$ T cell counts exceeded 400 cells/mm$^3$, and INRs as those whose CD4$^+$ T cell counts remained below 400 cells/mm$^3$ despite at least 2 years of ART with sustained virologic suppression. Among the PLWH cohort, 24 individuals met the criteria for INRs and 13 were classified as IRs. A comparison of the clinical characteristics of INRs and IRs is shown in Table S2.

We found that brassicasterol and campesterol levels were significantly higher in INRs (Wilcoxon rank-sum test, $P < 0.05$, Fig. 3A), whereas lanosterol levels were elevated in IRs (Wilcoxon rank-sum test, $P < 0.05$, Fig. 3A). Using Spearman's correlation coefficient ($\rho$), we found that the plant sterols brassicasterol and campesterol were negatively correlated with CD4$^+$ T cell counts as well as with NK cell counts, whereas lanosterol, a cholesterol precursor, was often positively correlated with CD4$^+$ T cell counts in PLWH (Fig. 3B and C).

To evaluate intergroup differences in intestinal cholesterol metabolism, we analyzed the relative abundance of *Oscillibacter* MSPs in IRs and INRs. Strikingly, INRs exhibited significantly higher relative abundances of *Oscillibacter* MSPs than IRs (Fig. S4), suggesting an enhanced microbe-mediated plant sterol absorption in the guts of INRs. This dysregulation may mechanistically contribute to impaired CD4$^+$ T cell recovery, as elevated plant sterol levels have been implicated in suppressing lymphocyte proliferation and differentiation.

Consistent with this hypothesis, we found that the abundance of *Oscillibacter* MSPs had a robust negative correlation with peripheral CD4$^+$ T cell counts in PLWH (Fig. S5). Notably, despite these functional differences, a comparative KEGG pathway analysis revealed the absence of any significant divergence in the overall metabolic potential between IRs and INRs (Fig. S6), underscoring the specificity of cholesterol metabolism alterations in driving immunological outcomes. Projection-based integrative analysis revealed that the increased abundance of *Oscillibacter* species was primarily associated with elevated levels of cholesterol metabolites and inversely correlated with plant sterol levels (Fig. 3D). Conversely, we observed a positive correlation between elevated cholesterol precursor levels and increased CD4$^+$ T cell counts, and a negative correlation between plant sterols and CD4$^+$ T cell recovery.

Given the significant correlations between the abundances of lanosterol, brassicasterol, and campesterol with the post-ART CD4$^+$ T cell recovery ($P < 0.05$), we further evaluated their predictive potential by developing a random forest classifier based on their abundances (Fig. 3E). The model was trained and tested using fivefold cross-validation to discriminate between IRs and INRs, and its diagnostic performance was validated by an ROC curve analysis. The results demonstrated that cholesterol did not have any predictive capability ($P > 0.05$), while brassicasterol exhibited the strongest predictive capability, achieving an AUC of 0.81 (95% CI = 0.67–0.95), significantly higher than those of lanosterol (AUC = 0.25) and campesterol (AUC = 0.71) (Fig. 3F). Hence, it can be suggested that brassicasterol can be used as a key biomarker for stratifying heterogeneous immune responses.

## Non-cholesterol sterols promote CD4$^+$ T cell recovery *in vitro*

To validate the functional effects of brassicasterol, campesterol, and lathosterol on CD4$^+$ T cell recovery, we conducted *in vitro* experiments by supplementing PBMCs obtained from HIV-infected individuals with these metabolites. Using the CCK-8 assay, we determined that 40 µM each of brassicasterol, campesterol, lathosterol, and cholesterol maintained relatively high viability of PBMCs from PLWH. Based on these findings, 40 µM was selected for further functional analysis (Fig. S7).

Flow cytometry was employed to evaluate the effects of treatment on various T cell subsets, a metric commonly used to assess HIV progression or prognosis. HIV infection induces persistent immune activation and inflammation, which can drive immune senescence and impair the ability of the immune system to effectively respond to pathogens or cancer cells, thereby exacerbating the prognosis of HIV-infected individuals (54). A hallmark of poor immune recovery in PLWH is the enrichment of effector T cell

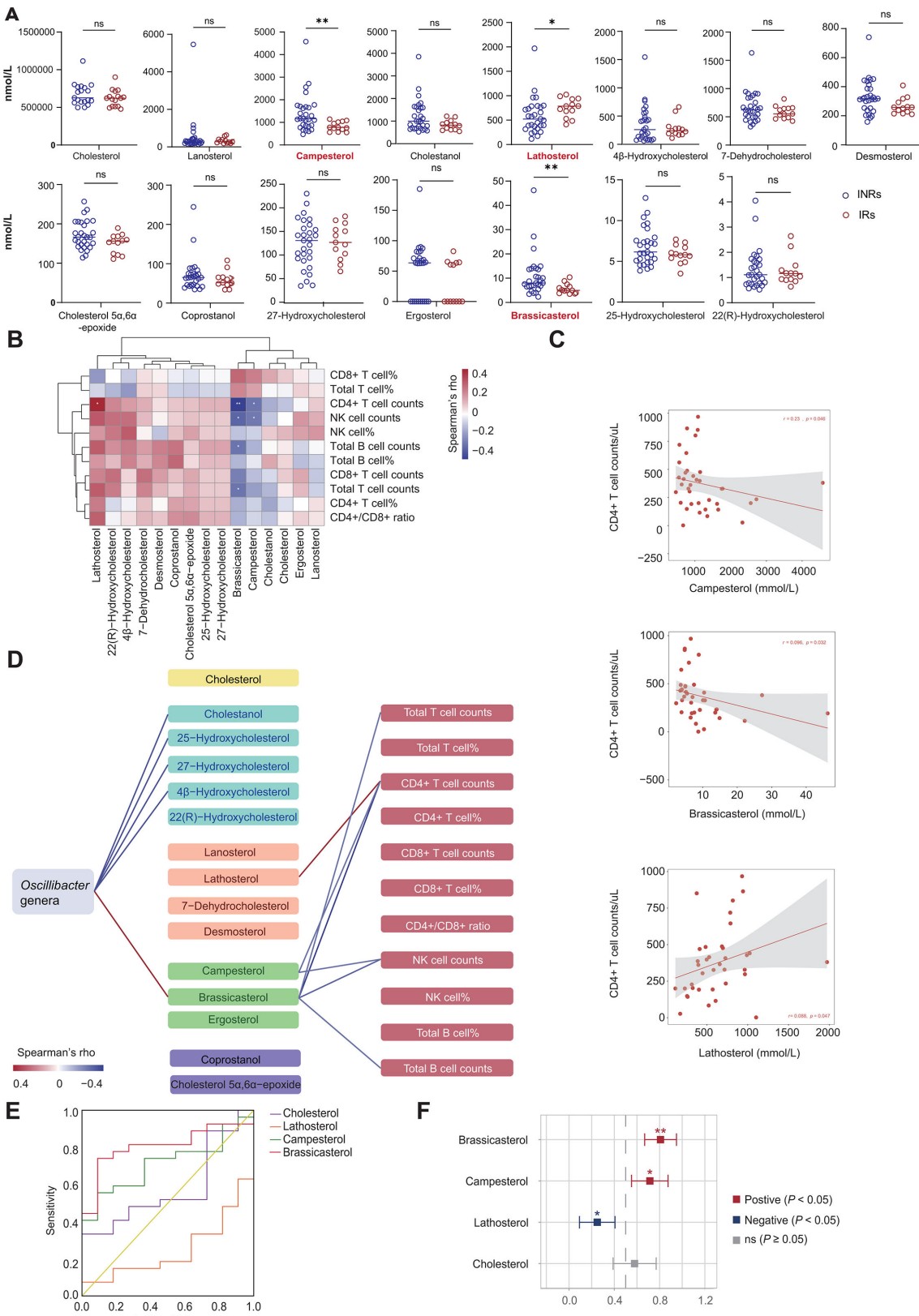

**FIG 3** Serum levels of non-cholesterol sterols and cholesterol in immunological non-responders (INRs) versus responders (IRs) and their correlations to post-antiretroviral therapy (ART) immune recovery. (A) Serum levels of non-cholesterol sterols and cholesterol in INRs versus IRs. Data presented as median ± IQR; *$P$ < 0.05, **$P$ < 0.01 by Mann–Whitney $U$-test. (B) Correlation heatmap between serum non-cholesterol sterols and cholesterol levels and immune cell

Fig 3 (Continued)

subset in PLWH. Color intensity reflects Spearman's ρ ($q$ value < 0.2, Benjamini–Hochberg corrected); asterisks denote significant correlations (*$P < 0.05$, **$P < 0.01$). (C) Scatterplots depicting the relationship between serum brassicasterol (left), campesterol (middle), and lathosterol (right) levels and CD4+ T cell counts in PLWH. Regression lines (solid) with 95% confidence intervals (CIs) (shaded). (D) Spearman's correlation analysis (based on projection-based integrative analysis) between the abundance of *Oscillibacter* species, serum sterol levels, and immune cell subsets in PLWH. In the middle panel, the yellow rounded rectangle represents cholesterol, blue indicates cholesterol metabolites, orange denotes cholesterol precursors, green represents plant sterols, and purple corresponds to other sterols. Red and blue lines indicate significant positive and negative Spearman's correlations, respectively ($P < 0.05$). (E and F) Differential markers for INRs and IRs in PLWH. (E) ROC curves for differential metabolites for discriminating INRs. (F) Areas under the ROC curves (95% CIs) of models based on individual biomarkers for the INRs ($n = 24$) versus the IRs ($n = 13$) (*$P < 0.05$, **$P < 0.01$).

phenotypes. These T cells typically exhibit heightened levels of activation, exhaustion, and senescence markers, including CD69, CD57, HLA-DR, and PD-1 (55, 56). Additionally, effector CD4+ T cells, naïve CD4+ T cells, and CD8+ T cells were analyzed as indicators of CD4+ T cell recovery in ART-treated PLWH (51). The levels of pro-inflammatory cytokines, such as tumor necrosis factor alpha (TNF-α), interleukin 6 (IL-6), and interleukin 1 beta (IL-1β), which drive chronic inflammation in plasma and lymph nodes, remain elevated from early infection stages and persist even in ART-treated individuals. Chronic inflammation, driven by these cytokines, is strongly associated with several negative outcomes and is ultimately associated with the clinical progression of HIV, including poor CD4+ T cell recovery and the onset of non-AIDS-related comorbidities (57).

Our results showed that treatment with brassinosteroid and campesterol significantly increased the proportions of HLA-DR+CD4+, CD57+CD4+, CD69+CD8+, and naïve CD8+ T cells and decreased those of effector CD4+ T cells. In contrast, treatment with lathosterol significantly increased the proportions of the effector CD4+ and CD8+ T cells and decreased those of PD1+CD4+ T cells. Cholesterol had a minimal impact on the T cell subset, with a significant reduction only in the proportions of PD1+CD4+ T and PD1+CD8+ T cells ($P < 0.01$) (Fig. 4A; Fig. S8).

To investigate the immunomodulatory effects of sterols in HIV pathogenesis, we analyzed cytokine production patterns through in vitro-stimulated PBMCs from PLWH. Consistent with our clinical observations, brassicasterol stimulation elicited a distinct pro-inflammatory cytokine profile compared to that in the PBS control, characterized by elevated production of IL-1β ($P < 0.05$), IL-2 ($P < 0.05$), IL-4 ($P < 0.05$), IL-6 ($P < 0.05$), IL-10 ($P < 0.05$), IL-13 ($P < 0.05$), and TNF-α ($P < 0.01$). Campesterol stimulation specifically enhanced IL-4 ($P < 0.01$), IL-6 ($P < 0.05$), and TNF-α ($P < 0.01$) levels, whereas lathosterol significantly attenuated IFN-γ production ($P < 0.05$). In contrast to non-cholesterol sterols, cholesterol had a minimal effect on the pro-inflammatory cytokine profile, with only a significant reduction in IL-8 production ($P < 0.05$) (Fig. 4B).

Complementary phenotypic analyses revealed that exposure to brassicasterol and campesterol promoted CD4+ T cell activation, exhaustion, and senescence and secretion of pro-inflammatory cytokines. Conversely, lathosterol decreased the secretion of pro-inflammatory cytokines without inducing exhaustion markers, suggesting a differential regulatory mechanism.

## DISCUSSION

This study investigated the intricate interplay between gut microbiome composition, non-cholesterol sterol metabolism, and immune reconstitution dynamics in PLWH. Our findings revealed significant alterations in the distribution of non-cholesterol sterols among PLWH, highlighting an association between specific *Oscillibacter* MSPs and non-cholesterol sterol profiles in this population. Notably, we identified plant sterols as potential contributors to impaired recovery of CD4+ T cells during ART. In contrast, lathosterol, a cholesterol precursor, was identified as a potential enhancer of CD4+ T cell restoration in PLWH. These observations were further validated through rigorous in vitro experiments, which confirmed the association between specific sterol metabolites and CD4+ T cell recovery during ART.

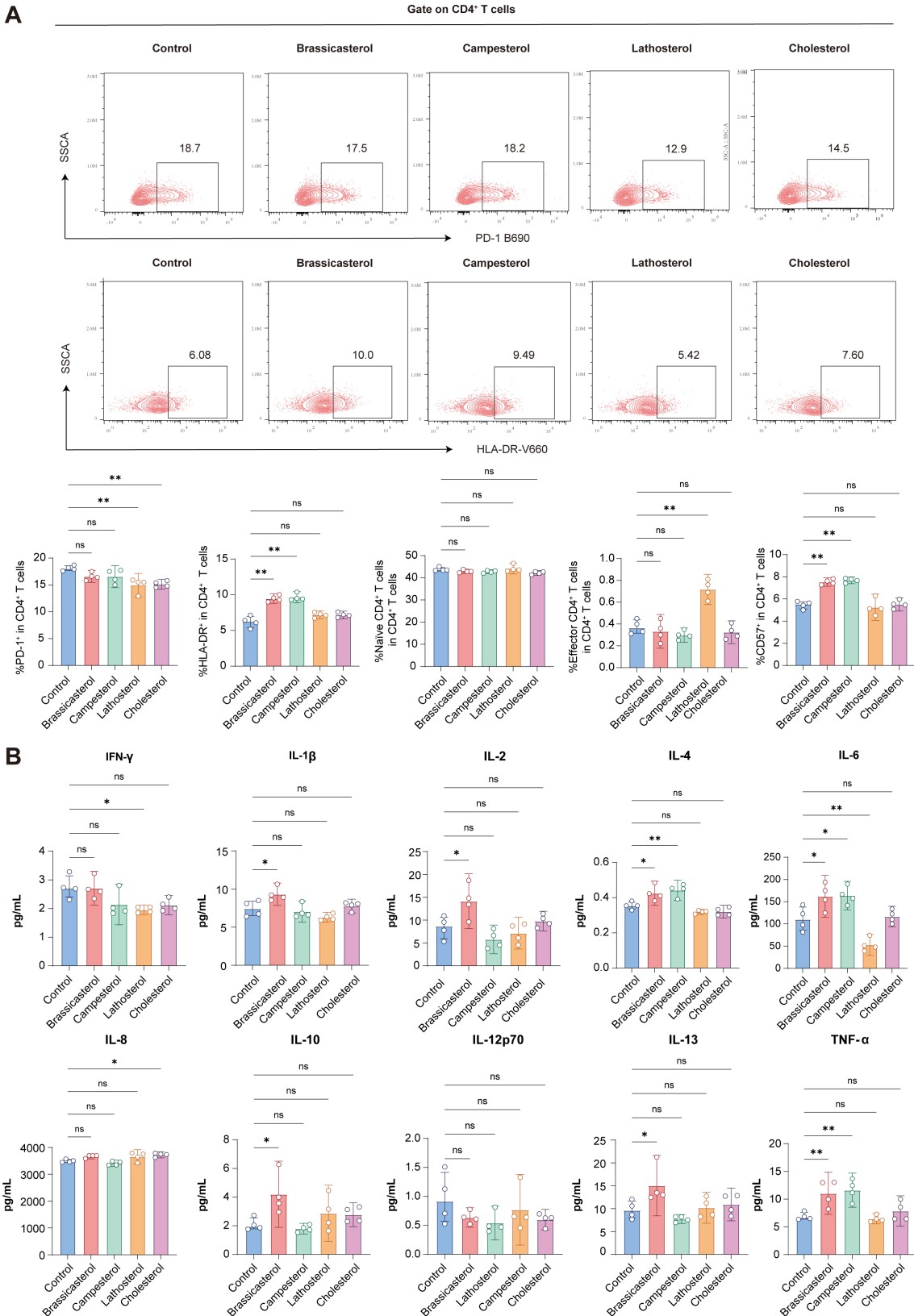

**FIG 4** Effect of non-cholesterol sterols and cholesterol stimulation on immune markers in peripheral blood mononuclear cells (PBMCs). (A) Flow cytometry analysis of T cell markers following 48 h stimulation with 40 µM of brassicasterol, campesterol, lathosterol, or cholesterol ($n = 4$ donors; mean ± SEM; *$P < 0.05$, **$P < 0.01$ by one-way ANOVA with Tukey's post hoc test). (B) Meso-scale discovery (MSD) multiplex profiling of pro-inflammatory cytokines following 48 h stimulation with 40 µM of brassicasterol, campesterol, lathosterol, or cholesterol ($n = 4$ donors; mean ± SEM; *$P < 0.05$, **$P < 0.01$ by one-way ANOVA with Tukey's post hoc test).

A previous study reported that long-term ART in patients with HIV is associated with significant alterations in their lipid profiles, often leading to dyslipidemia (58). Furthermore, emerging evidence suggests that disease progression in HIV-infected individuals may be delayed in those exhibiting enhanced cholesterol metabolism and reduced cholesterol synthesis (59, 60). Cholesterol removal has been shown to prevent HIV-induced syncytium formation *in vitro*, lower the buoyant density of viral particles, disrupt co-receptor expression, and enhance cell resistance to infection, thereby leading to a substantial reduction in viral infectivity. However, various recent studies suggest that hypocholesterolemia in PLWH may be associated with unfavorable immune reconstitution, which highlights the need for further investigation into the balance between cholesterol modulation and immune function in this population (61, 62).

Our study is a comprehensive investigation into cholesterol metabolism, synthesis, and absorption and reveals significant alterations in these metabolic pathways among PLWH compared to those in HCs. Specifically, PLWH exhibited elevated levels of cholesterol precursors, such as lanosterol, and cholesterol metabolites, including cholestanol, 4β-hydroxycholesterol, 27-hydroxycholesterol, and 25-hydroxycholesterol. Conversely, the levels of plant sterols, such as brassicasterol and ergosterol, were markedly reduced in PLWH relative to those in HC. These findings underscore the distinct patterns of cholesterol biosynthesis and absorption in PLWH, which were closely associated with gut dysbiosis in this population. The gut microbiota exhibits diverse immunomodulatory functions, which are attributed to variations in their antigenicity, microbial aggressiveness, and the metabolites they produce (63). Notably, *Oscillibacter* MSPs, previously identified for their ability to metabolize cholesterol, were found to have significantly reduced relative abundance among PLWH. The relative abundance of *Oscillibacter* MSPs was negatively correlated with cholesterol metabolites and precursors and positively with plant sterols. Hence, it can be suggested that *Oscillibacter* MSPs may help cholesterol degradation and plant sterol absorption in the intestine. Notably, in the gut microbiome of PLWH, KEGG pathways related to steroid degradation and steroid hormone biosynthesis were diminished, whereas pathways associated with steroid biosynthesis were enriched. These findings suggest that *Oscillibacter* may play a key role in modulating cholesterol metabolism in PLWH.

As the primary target cells of HIV, CD4$^+$ T cells play a pivotal role in the pathogenesis of the disease and are directly linked to clinical prognosis (4–6). The progressive depletion of CD4$^+$ T cells not only signifies immune deterioration but also serves as a critical marker for treatment outcomes and susceptibility to opportunistic infections in PLWH (8). Therefore, understanding the factors that influence CD4$^+$ T cell recovery during ART is essential for improving long-term prognosis. Recent studies have highlighted the gut microbiota as an important determinant of immune reconstitution in PLWH (52, 57). Among them, the lactic acid- and SCFA-producing species are related to immune recovery in PLWH (57). Elevated SCFAs and lactate create an immunopromotive effect or prevent inflammation and reduce the subsequent morbidity and mortality (43, 44). Our study showed that (i) the *Oscillibacter* species were positively associated with brassicasterol, (ii) brassicasterol and campesterol levels were higher in INRs, (iii) lanosterol levels were elevated in IRs, (iv) brassicasterol and campesterol were negatively associated with CD4$^+$ T cell counts, and (v) lanosterol was positively associated with CD4$^+$ T cell counts in PLWH. These observations suggest that the *Oscillibacter* species and the sterols identified in this study underwent significant modifications, which points to their potential role in shaping diverse immune responses to ART. We therefore suggest that elevated serum lathosterol concentrations contribute to the immunopromotive effect, while brassicasterol levels will deplete the immune recovery in PLWH. These elements may represent potential targets for further exploration into the mechanisms driving immune recovery in PLWH.

To confirm the impact of brassicasterol, campesterol, and lathosterol on the CD4$^+$ T cell recovery, we evaluated their immunomodulatory effects using *ex vivo* PBMC models. Chronic systemic T cell activation is a hallmark of incomplete immune recovery in PLWH,

strongly correlating with impaired CD4$^+$ T cell reconstitution and serving as an independent predictor of adverse long-term clinical outcomes (5, 64). Elevated circulating inflammatory markers—particularly IL-6 and D-dimers—further predict morbidity and mortality during cART (64–66). Higher levels of canonical pro-inflammatory cytokines, including IL-1β, IL-6, and IL-8, are associated with improved CD4$^+$ T cell recovery at month 12, but are linked to a decrease in CD4$^+$ T cell recovery (57). Notably, brassicasterol and campesterol induce systemic T cell activation and amplify pro-inflammatory cytokine production, with brassicasterol exhibiting the most pronounced effects. In stark contrast, lathosterol suppresses both T cell activation and inflammatory responses. Cholesterol elicits context-dependent effects, predominantly demonstrating immunoregulatory benefits but showing paradoxical pro-inflammatory activity in specific subsets.

Our results reveal the disorder in cholesterol and non-cholesterol sterol metabolism in PLWH, indicating the importance of the gut microbiome as a modulator of host sterol metabolism in HIV infection. The association between *Oscillibacter* MSPs and non-cholesterol sterols suggests that microbial-driven metabolic pathways may influence systemic immune responses, particularly in PLWH. The detrimental impact of plant sterols on CD4$^+$ T cell recovery aligns with the emerging evidence linking dysregulated cholesterol and non-cholesterol sterol metabolism to immune dysfunction in HIV infections. Conversely, the positive correlation between lathosterol and CD4$^+$ T cell restoration highlights a potential therapeutic avenue for enhancing immune recovery during ART.

These findings have broader implications for understanding the role of microbial–host metabolic interactions in chronic diseases. By elucidating the mechanisms underlying sterol-mediated immune modulation, our study provides a foundation for future research aimed at developing microbiome-targeted therapies to improve clinical outcomes in PLWH. Further investigation into the specific pathways by which *Oscillibacter* species and non-cholesterol sterols influence immune function will be critical for translating these findings into actionable therapeutic strategies.

Our study has several limitations. First, the sample size is relatively small, and thus, our findings need to be validated in larger cohorts representing diverse ethnicities. Second, while the study controlled for various potential confounders influencing the gut microbiome, our cohort lacked data on factors that may also affect the microbiome and sterol metabolism, such as medication, diet, and other health conditions in the participants. Third, the dynamic changes in the gut microbiome, non-cholesterol sterols, and cholesterol metabolism throughout the course of HIV infection were not assessed. Fourth, the potential impact of co-infections on the gut microbiota was not accounted for in our analysis, which may have introduced a degree of bias into the results. Finally, further *in vitro* experiments are needed to provide additional mechanistic insights into the effects of bacteria on immune cell function.

## Conclusion

This study underscores the intricate relationship between gut microbiota composition, non-cholesterol sterol metabolism, and immune reconstitution in PLWH. The study demonstrates that gut dysbiosis is associated with the dysregulation of sterol metabolism in PLWH. Plant sterols (brassicasterol and campesterol) impede CD4$^+$ T cell recovery, while host-adapted cholesterol derivatives (e.g., lathosterol) promote sustainable immune restoration. Crucially, the depletion of *Oscillibacter* species, microbial architects of sterol catabolism, is a biomarker as well as a therapeutic target for ART optimization. By mechanistically linking microbial cholesterol degradation capacity to systemic immunometabolic homeostasis, our findings advocate for precision strategies that recalibrate gut microbial functions—through targeted enrichment of sterol-modulating taxa or dietary sterol modulation—to resolve chronic inflammation while preserving adaptive immune resilience. This paradigm extends beyond HIV management, offering a template for microbiome-metabolite engineering in chronic inflammatory diseases.

## ACKNOWLEDGMENTS

This study was funded by the National Key R&D Program of China (grant number 2024YFC2309900), the Zhejiang Plan for the Special Support for Top-notch Talents (grant number 2022R52029), the Fundamental Research Funds for the Central Universities (grant number 2022ZFJH003), and Shandong Provincial Laboratory Project (grant number SYS202202).

H.Y. and L.L. designed and supervised the study. J.P., X.T., M.D., and P.Y. recruit participants and collect samples. J.P., K.W., J.J., T.S., and D.L. performed experiments and analyzed the data. J.P. and X.T. drafted the manuscript. All authors revised the manuscript and approved the final version.

## AUTHOR AFFILIATIONS

[1]State Key Laboratory for Diagnosis and Treatment of Infectious Diseases, National Clinical Research Center for Infectious Diseases, Collaborative Innovation Center for Diagnosis and Treatment of Infectious Diseases, The First Affiliated Hospital, School of Medicine, Zhejiang University, Hangzhou, China

[2]Jinan Microecological Biomedicine Shandong Laboratory, Jinan, China

[3]Department of Infectious Diseases, Zhejiang Qingchun Hospital, Zhejiang University, Hangzhou, China

[4]Wenzhou Collaborative Innovation Center of Gastrointestinal Cancer in Basic Research and Precision Medicine, Wenzhou Key Laboratory of Cancer-related Pathogens and Immunity, Institute of Tropical Medicine, School of Basic Medical Sciences, Wenzhou Medical University, Wenzhou, China

## AUTHOR ORCIDs

Jingying Pan http://orcid.org/0000-0002-6804-7951
Xuebin Tian http://orcid.org/0000-0003-0207-6917
Longxian Lv http://orcid.org/0000-0003-0204-5862
Hangping Yao http://orcid.org/0000-0001-6742-7074

## FUNDING

| Funder | Grant(s) | Author(s) |
| --- | --- | --- |
| National Key Research and Development Program of China | 2024YFC2309900 | Hangping Yao |
| Zhejiang Plan for the Special Support fort Top-notch Talents | 2022R52029 | Hangping Yao |
| Fundamental Research Funds for the Central Universities | 2022ZFJH003 | Hangping Yao |
| Shandong Provincial Laboratory Project | SYS202202 | Hangping Yao |

## AUTHOR CONTRIBUTIONS

Jingying Pan, Conceptualization, Data curation, Formal analysis, Resources, Software, Validation, Visualization, Writing – original draft, Writing – review and editing | Xuebin Tian, Data curation, Investigation, Methodology, Software, Writing – review and editing | Kai Wu, Data curation, Validation, Visualization, Writing – original draft, Writing – review and editing | Jia Ji, Data curation, Formal analysis, Validation, Visualization | Mingqing Dong, Data curation, Methodology | Ting Sun, Data curation, Methodology, Validation, Visualization | Dan Lv, Data curation, Methodology | Peng Yao, Data curation, Methodology | Longxian Lv, Conceptualization, Formal analysis, Methodology, Supervision, Validation, Writing – review and editing.

## DATA AVAILABILITY

The raw metagenomic sequencing data generated in this study have been deposited in the National Genomics Data Center (https://ngdc.cncb.ac.cn/gsa/) under accession no. PRJCA021055. Additional data analyzed during the study are available from the corresponding author upon reasonable request.

## ETHICS APPROVAL

This study was approved by the Institutional Review Board of the First Affiliated Hospital, Zhejiang University School of Medicine (approval no. IIT20230314B), with written informed consent obtained from all participants.

## ADDITIONAL FILES

The following material is available online.

### Supplemental Material

**Supplemental figures and tables (Spectrum01404-25-s0001.docx).** Fig. S1 to S8, and Tables S1 and S2.

### Open Peer Review

**PEER REVIEW HISTORY (review-history.pdf).** An accounting of the reviewer comments and feedback.

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
