## [Reviewer comments · Microbiology Spectrum]

Microbiology Spectrum

Gut microbiota-associated non-cholesterol sterol dysregulation modulates immune reconstitution during antiretroviral therapy in people living with HIV

Jingying Pan, Xuebin Tian, Kai Wu, Jia Ji, Mingqing Dong, Ting Sun, Dan Lv, Peng Yao, Longxian Lv, and Hangping Yao

Corresponding Author(s): Hangping Yao, Zhejiang University School of Medicine First Affiliated Hospital State Key Laboratory For Diagnosis and Treatment of Infectious Diseases

Review Timeline:

Submission Date:	May 8, 2025
Editorial Decision:	June 5, 2025
Revision Received:	June 18, 2025
Accepted:	June 20, 2025

Editor: Benjamin Liu

Reviewer(s): Disclosure of reviewer identity is with reference to reviewer comments included in decision letter(s). The following individuals involved in review of your submission have agreed to reveal their identity: AJAY KUMAR AJAY KUMAR (Reviewer #1); Shahid Siddhik Attar (Reviewer #2)

Transaction Report:

DOI: <https://doi.org/10.1128/spectrum.01404-25>

Re: Spectrum01404-25 (**Gut microbiota-associated non-cholesterol sterol dysregulation modulates immune reconstitution during antiretroviral therapy in people living with HIV**)

Dear Prof. Hangping Yao:

Thank you for the privilege of reviewing your work. Below you will find my comments, instructions from the Spectrum editorial office, and the reviewer comments.

Editor's comments:

1. The authors should introduce the role of CD4+ T cells in anti-HIV functions. More references should be cited, with this one (PMID: 24191027) as an example (citing is optional).
2. The authors should introduce that PLWH tend to have co-infection with other pathogens, e.g. mpox, STD, etc. The authors should discuss the difference between colonized microbiota and co-infection in PLWH. More references should be cited, with this one (PMID: 38793665) as an example (citing is optional).

Please return the manuscript within 30 days; if you cannot complete the modification within this time period, please contact me. If you do not wish to modify the manuscript and prefer to submit it to another journal, notify me immediately so that the manuscript may be formally withdrawn from consideration by Spectrum.

Revision Guidelines

Sincerely,
Benjamin Liu

Editor
Microbiology Spectrum

Reviewer #1 (Comments for the Author):

Gut microbiota-associated non-cholesterol sterol dysregulation modulates immune reconstitution during antiretroviral therapy in people living with HIV must go or wet lab analysis for better ngs sequence

Reviewer #2 (Comments for the Author):

This article is important for future study, CD4+ T-cell recovery is need in HIV patient.

Dear Editors and reviewers:

Thank you for giving us the opportunity to submit a revised draft of my manuscript titled " Gut microbiota-associated non-cholesterol sterol dysregulation modulates immune reconstitution during antiretroviral therapy in people living with HIV" (Spectrum01404-25) to *Microbiology Spectrum*. We appreciate the time and effort that you have dedicated to providing your valuable feedback on my manuscript. We are grateful to you for your insightful comments on my paper. According to the associate editor and reviewers' comments, we have made extensive modifications to our manuscript and supplemented extra data to make our results convincing. Thank you again for your positive comments and valuable suggestions to improve the quality of our manuscript.

Here is a point-by-point response to the editor and reviewers' comments and concerns.

Editor's comments:

1. The authors should introduce the role of CD4⁺ T cells in anti-HIV functions. More references should be cited, with this one (PMID: 24191027) as an example (citing is optional).

Response: Thank you for this insightful suggestion. We fully agree that the role of CD4⁺ T cells in anti-HIV immune responses is critical and warrants clear discussion in the *Introduction*. In response, we have revised the relevant section to better clarify the dual role of CD4⁺ T cells in HIV infection—not only as primary targets of viral entry and replication, but also associated with clinical outcome.

Additionally, we have incorporated several relevant references, including the one suggested by the Editor (*Liu S, et al. MCP1 restricts HIV infection and is rapidly degraded in activated CD4+ T cells. Proc Natl Acad Sci U S A. 2013;110(47):19083-19088*; now cited as reference (4)), which provides important insights into CD4⁺ T cell-mediated immune control. These revisions have been implemented in the *Introduction* (page 4, lines 73–82) and *Discussion* (page 19, lines 503–509) of the revised manuscript.

2. The authors should introduce that PLWH tend to have co-infection with other pathogens, e.g. mpox, STD, etc. The authors should discuss the difference between colonized microbiota and co-infection in PLWH. More references should be cited, with this one (PMID: 38793665) as an example (citing is optional).

Response: We sincerely thank you for this valuable suggestion. In response, we have revised the *Introduction* to address the increased susceptibility of people living with HIV (PLWH) to co-infections with other pathogens such as mpox virus, sexually transmitted infections (STIs), hepatitis B virus (HBV), hepatitis C virus (HCV), and *Treponema pallidum*. We now emphasize that these co-infections can further exacerbate immune dysfunction and complicate disease management in PLWH. Relevant references have been added to support these statements, including the suggested article (Liu BM, et al. *Mpox (Monkeypox) Virus and Its Co-Infection with HIV, Sexually Transmitted Infections, or Bacterial Superinfections: Double Whammy or a New Prime Culprit? Viruses*. 2024;16(5):784) (now cited as reference 29 in the revised manuscript, page 5, lines 109–118).

Furthermore, as co-infections may influence the gut microbiome, we have acknowledged a limitation in our study: the potential impact of co-infections on gut microbial composition was not accounted for in our analysis, which may have introduced a degree of bias into the results. This has been addressed in the *Discussion* section (page 21, lines 560–561).

Reviewer #1 (Comments for the Author):

Gut microbiota-associated non-cholesterol sterol dysregulation modulates immune reconstitution during antiretroviral therapy in people living with HIV must go or wet lab analysis for better ngs sequence.

Response: Thank you for the insightful suggestion regarding the need for additional validation of the sequencing-based findings. We fully agree that wet-lab analysis is essential to support and enhance the interpretation of NGS data.

In this study, we have functionally validated the immunomodulatory role of non-cholesterol sterols through ex vivo stimulation assays using PBMCs isolated from PLWH. However, we

have not yet performed wet-lab experiments to directly investigate the functional role of *Oscillibacter spp.* in modulating host immunity. We appreciate the reviewer's point, and we acknowledge that such experiments would greatly strengthen the conclusions. In line with this, we are currently planning follow-up studies to explore the immunological impact and underlying mechanisms of *Oscillibacter spp.* in PLWH, which we expect will provide deeper insight into the microbiota–host interaction landscape.

Besides, for better ngs sequence analysis, we have expanded our analysis by integrating gut metagenomic data with clinical laboratory parameters. Notably, we observed that the relative abundance of *Oscillibacter spp.* was positively correlated with high-density lipoprotein cholesterol (HDL-C) and negatively correlated with triglyceride (TG) levels in PLWH. These results are in line with previous studies linking *Oscillibacter* to cholesterol metabolism and further support the hypothesis that microbiota-associated sterol dysregulation may contribute to immune reconstitution through cholesterol metabolism pathways. This new analysis has been added to the revised manuscript (Results section, pages 13-14, lines 350-356; Figure S3).

Reviewer #2 (Comments for the Author):

This article is important for future study, CD4⁺ T-cell recovery is need in HIV patient.

Response: We sincerely thank you for the positive and encouraging comments. We agree that CD4⁺ T-cell recovery is a critical aspect of HIV treatment and long-term prognosis. We have also emphasized the role of CD4⁺ T cells in anti-HIV immunity in the Introduction (page 4, lines 73–82) and Discussion (page 19, lines 503–509) of the revised manuscript. We hope that our findings contribute to a better understanding of the factors influencing immune reconstitution in people living with HIV and provide a basis for future mechanistic and therapeutic studies.

Once again, we sincerely thank you for your valuable comments and suggestions. We hope that the revised manuscript will meet the journal's standards and be considered for publication in *Microbiology Spectrum*.

Yours sincerely,

Hangping Yao

E-mail: yaohangping@zju.edu.cn

Re: Spectrum01404-25R1 (**Gut microbiota-associated non-cholesterol sterol dysregulation modulates immune reconstitution during antiretroviral therapy in people living with HIV**)

Dear Prof. Hangping Yao:

Your manuscript has been accepted, and I am forwarding it to the ASM production staff for publication. Your paper will first be checked to make sure all elements meet the technical requirements. ASM staff will contact you if anything needs to be revised before copyediting and production can begin. Otherwise, you will be notified when your proofs are ready to be viewed.

Sincerely,
Benjamin Liu
Editor
Microbiology Spectrum